# Successful Islet Outcomes Using Australia-Wide Donors: A National Centre Experience

**DOI:** 10.3390/metabo11060360

**Published:** 2021-06-05

**Authors:** Wayne J Hawthorne, Sussan Davies, Hee-chang Mun, Yi Vee Chew, Lindy Williams, Patricia Anderson, Natasha Rogers, Philip J O’Connell

**Affiliations:** 1The Centre for Transplant & Renal Research, Westmead Institute for Medical Research, Westmead Hospital, Westmead, NSW 2145, Australia; sussan.davies@sydney.edu.au (S.D.); hee.mun@sydney.edu.au (H.-c.M.); yi.chew@sydney.edu.au (Y.V.C.); lindy.williams@sydney.edu.au (L.W.); Patricia.Anderson@health.nsw.gov.au (P.A.); natasha.rogers@health.nsw.gov.au (N.R.); philip.oconnell@sydney.edu.au (P.J.O.); 2Westmead Clinical School, University of Sydney, Westmead Hospital, Westmead, NSW 2145, Australia

**Keywords:** diabetes, hypoglycemic unaware, ischaemia, islet cell transplantation, organ donation, pancreas

## Abstract

Cold ischemia and hence travel time can adversely affect outcomes of islet isolation. The aim of this study was to compare the isolation and transplant outcomes of donor pancreata according to the distance from islet isolation facility. Principally, those within a 50 km radius of the centre were compared with those from regional areas within the state and those from interstate donors within Australia. Organ donors were categorised according to distance from National Pancreas Transplant Unit Westmead (NPTU). Donor characteristics were analysed statistically against islet isolation outcomes. These were age, BMI, cause and mechanism of death, days in ICU, gender, inotrope and steroid use, cold ischemia time (CIT) and retrieval surgical team. Between March 2007 and December 2020, 297 islet isolations were performed at our centre. A total of 149 donor pancreata were local area, and 148 non-local regions. Mean distance from the isolation facility was 780.05 km. Mean pancreas CIT was 401.07 ± 137.71 min and was significantly different between local and non-local groups (297.2 vs. 487.5 min, *p* < 0.01). Mean age of donors was 45.22 years, mean BMI was 28.82, sex ratio was 48:52 F:M and mean time in ICU was 3.07 days. There was no significant difference between local and non-local for these characteristics. The mean CIT resulting in islet transplantation was 297.1 ± 91.5 min and longest CIT resulting in transplantation was 676 min. There was no significant difference in islet isolation outcomes between local and non-local donors for characteristics other than CIT. There was also no significant effect of distance from the isolation facility on positive islet transplant outcomes (C-peptide > 0.2 at 1 month post-transplant). *Conclusions:* Distance from the isolation centre did not impact on isolation or transplant outcomes supporting the ongoing nationwide use of shipping pancreata for islet isolation and transplantation.

## 1. Introduction

Pancreatic islet cell transplantation has become a successful modality of treatment for a select group of patients with type 1 diabetes. [1,2,3,4]. In order to provide this service, centralized islet isolation centres need to overcome a number of unique logistical problems, in particular retrieving donor pancreata and transplanting patients from distant areas. In Australia, the problem is particularly acute [5,6] as the service covers an area of 7,692,024 square kilometers, which is approximately twice the size of Europe or three-quarters the size of the United States. The National Islet Consortium comprises two isolation centres—Westmead Hospital in Sydney and St Vincent’s Hospital in Melbourne; and three transplantation Hospitals—Westmead, St Vincent’s and Queen Elizabeth in Adelaide. [6] Almost one-third of the Australian population lives outside these major urban centres and patients from regional and rural areas face a number of barriers to accessing medical services (Figure 1). The aim of this study was to assess whether pancreas donors accessed outside the local region of the isolation facility provided equivalent outcomes to those accessed from the local region of the isolation facility.

## 2. Results

### 2.1. The Effect of Distance on Isolation Outcome

Between March 2007 and December 2021, 297 islet isolations were performed and analysed for this study. Donor pancreata came from multiorgan donors that were accessed from multiple hospitals across Australia (149 were from the local region, 148 were from non-local regions). The mean distance from isolation centre was 780.05 km (range 0 to 3278.7 km) for all donors. The distance from the isolation facility provided significant impact on the islet isolation outcome the closer local donors provided a positive outcome at 11.34 km ± 10.22 vs. 883.04 km ± 625.45 for a positive outcome from non-local donors, *p* =< 0.01) (Figure 2A). The chance of obtaining an isolation outcome that achieved release criteria was greater from local donors with a positive outcome occurring 32% of the time from local donors as compared to only 24% of isolations from non-local donors (*p* < 0.01).

The mean pancreas cold ischemic time for all donors was 401.07 ± 137.71 min (range 78 to 870). CIT correlated closely with distance from isolation facility (Figure 2B). From local (zone 1) to zone 2, the mean CIT increased from 279.6 ± 89.2 to 337.6 ± 74.4 min (*p* = 0.011). As one moves out from zone 2 to zone 6, there was a progressive increase in cold ischemia times. (The mean CIT for each zone is shown on Table 1.) When all non-local donors were compared, there was a significant increase in mean CIT when compared to local CIT (mean 487.5 ± 103.4 vs. 297.2 ± 95.7 min, Figure 2C, *p* = 0.12). However, regardless of the region, there was a negative association between CIT and chances of a positive isolation outcome (*p* = 0.55). Although isolations with a positive outcome tended to have a shorter CIT compared to those with a negative outcome (297.1 ± 91.5 min vs. 297.6 ± 105.5 min) (Figure 2C) this was not statistically significant (*p* = 0.73) (Figure 2C).

### 2.2. Other Donor Factors Affecting Isolation Outcome

Forty-eight percent of the donors were female and patient gender had no effect on isolation outcomes (*p* = 0.554) and there were no differences in the gender balance between local and non-local donors (*p* = 0.6493, data not shown). There was no difference in age between donors accepted from local versus those from non-local regions (44.5 ± 12.3 vs. 45.7 ± 12.2, *p* = 0.46). The mean age of all donors was 45.22 years (range 10 to 71), and there was no difference in age of the donor between those with a positive isolation outcome and those that did not (44.34 (range 23 to 69) vs. 45.49 (range 14 to 71), *p* = 0.46, Figure 3A). The mean BMI for all donors was 28.82 kg/m^2^ (range 19.80 to 57.46) and the mean BMI was greater in isolations with a positive isolation outcome with those that did not (31.01 (range 20.09 to 51.05) vs. 28.12 (range 17.90 to 57.46), *p* = 0.0004, Figure 3B). However, there was no difference in the BMI between local compared with non-local groups (28.81 (range 17.48 to 50.50) vs. 28.90 (range 19.81 to 57.46), *p* = 0.927). The mean time spent in ICU for all donors was 3.07 days (range 1 to 13) and no impact on isolation outcome, (mean time for a positive outcome 2.8 ± 1.7 days versus 3.1 ± 2.7 days for a negative outcome and this was not significantly different (*p* = 0.451, Figure 3C), nor was it significant between local and non-local groups (2.85 days vs. 3.10 days). Cause and mechanism of death demonstrated that ‘other causes’ or ‘other mechanisms’ have the highest positive results with a significant *p* value (*p*= 0.042) (Table 2), whereas inotrope dose, steroid use and retrieval team had no significant effect on the islet isolation outcome (Table 1).

The clinical outcome of those isolations that were transplanted were assessed and evaluated for their effect of donor-related factor on outcome using the non-parametric Spearman’s rank correlation. Donor distance, CIT, days in ICU, donor BMI had no effect on C-peptide levels achieved (Table 3).

## 3. Discussion

Availability of suitable donor pancreata is a major limiting factor for islet transplant activity. Hence, the ability to access pancreata that are retrieved at a distance from the isolation centre is essential if more patients are to be transplanted. However, it is essential to ascertain whether pancreata retrieved from more distant centres provide equivalent outcomes to those retrieved locally. This study evaluated the differences in donor characteristics between locally retrieved local donor organs with those retrieved from non-local regions. Because of the distances between major cities and states within Australia there was a large variation in distances between the donor hospital and the isolation centre. Because of the logistics of transport, there were substantial differences in CIT between local and non-local donors. This did impact on the chances of achieving a positive isolation resulting in reaching release criteria. However, provided the release criteria were met there were no differences in transplant outcomes between local and non-local donors. The one caveat to this conclusion was that no successful isolation was achieved if the CIT was greater than 676 min.

The contributing factors that affect the logistics of the transport of the organ included- time from organ retrieval to shipping. These included the time to separate the pancreas from the liver after removal en-bloc during organ retrieval and the availability of a commercial flight to Sydney. These variables tended to have a substantial impact on CIT. These logistical issues had a major impact on organs retrieved from regions 3 to 5 where flight times range from 44 min to 2 h. Pancreata transported from zone 6 had flight times ranging from 3 to 6 h. By contrast those retrieved within the local region were transported to the isolation facility by car which meant that it tended to travel with the retrieval team and did not suffer from delays caused by transfer to couriers and delays with airport transfers. In most instances though the careful planning of organ donor retrieval surgical times around flight times and urgency to meet assigned flights made significant differences to flight times as seen between zone 6 and those undertaken in zone 4 where similar urgency to make flight times was not as critical or there were significant distances driven from a distant regional hospital to make the flight interstate.

Other studies have looked at the impact of distance on isolation outcome including an initial report of 3 patients with a follow up of 11 patients, transplanted with islets isolated from pancreata obtained in Houston that were transported by air to the isolation centre in Miami, before being transported back to be implanted into patients in Houston [7,8,9]. However, the major focus was on recipient outcomes with transplanted preparations and the analysis of the factors affecting isolation were limited. More recently, established consortiums have been actively recruiting remote sites, with the UK and GRAGIL being notable examples [10]. In the GRAGIL consortium, pancreata are transported to a single isolation centre in Geneva and the distances travelled were all within 300 min driving time.

The study reported by the GRAGIL study is a relatively small number of isolations performed and the consequent small number of preparations suitable for transplantation. Whilst we did not identify an impact of CIT and days in ICU on isolation outcome, these have been identified as important variables in other studies. In this setting, CIT is a good surrogate for distance travelled but only when taken in the right context and in relation to the exceptionally long distances travelled across Australia that can only occur by air travel. CIT has been shown to have an adverse outcome on islet isolation in both single-centre and registry studies [1,9].

A unique feature of this study was the distance travelled between the donor hospital and the islet isolation facility. Donor pancreata were transported up to 3290 km from the city of Perth situated on the west coast to Sydney on the east coast of Australia for islet isolation. Whilst these large distances did impact on the chances of a successful outcome, we show that it is possible to achieve release criteria and good outcomes from those organs that come from distant regions.

The BMI of a donor was also significant in determining a positive isolation outcome meeting release criteria and thus we must continue to consider this when deciding whether we should perform a donor pancreas isolation. The BMI of a donor has been found to correlate with the size of their pancreas and bigger pancreata often result in greater islet numbers [11]. This result is consistent with other islet groups’ isolation and transplant outcomes around the world [12,13].

All other factors were not significantly different between all comparison groups. The data we used for this analysis were retrospective and donor selection criteria were based on the Edmonton Score [14]. If possible, we choose organs that are in an accepted age group, larger BMI, limited hypoxia and minimal or no steroid use as these things have been shown to impact on isolation results [11,14,15,16]. This may have prevented us identifying factors other than distance and donor BMI as important criteria for achieving release criteria.

In conclusion, excellent islet transplant outcomes can be achieved from pancreata retrieved at distant centres, despite substantial logistical issues involved. The nationally funded program provides a fair and equitable use for donor organs regardless of the state they are retrieved in and will provide outcomes for patients equivalent to the best units in the world despite major logistical hurdles compared to other units. Whilst pancreata retrieved from distant sites are less likely to achieve release criteria, those that do achieve release criteria have comparable outcomes to locally retrieved outcomes. Distant pancreas retrieval does pose challenges and careful selection of appropriate donors and minimisation of CIT by improved logistics is essential to ensure success.

## 4. Materials and Methods

### 4.1. Donor Selection

All multiorgan heart-beating brain dead donors were accepted for donation based upon their suitability as described previously [15,17]. Organ donor characteristics that could influence islet isolation and transplantation outcomes were recorded including cold ischemia time (CIT), transport time, donor age, BMI, admission blood glucose levels, hypotension, use of vasopressors prior to death, and cause of death [18].

### 4.2. Islet Preparation

Islets were separated as described previously using a variation of the closed-loop method described by Ricordi et al. [5,18]. Pancreata were disaggregated by infusing the ducts with cold collagenase NB1 GMP grade (SERVA, Heidelberg, Germany). Dissociated islet and acinar tissue were separated on a continuous Biocoll (Biochrom AG, Berlin, Germany) density gradient (polysucrose 400 and amidotrizoic acid) on a refrigerated apheresis system (Model 2991, COBE Laboratories, Lakewood, Colorado).

### 4.3. Release Criteria

Purified islets were counted and islet number and mass were expressed in terms of islet equivalents (IEQ) [19]. Islet preparations underwent pre-culture quality assurance, which included purity and viability assessment, packed cell volume measurement and evaluation of islet morphology to exclude excessive fragmentation. Islets were cultured in Miami media in 95% room air and 5% CO_2_ at 37 °C for up to 24 h with quality assurance, including beta cell viability index, oxygen consumption rate, endotoxin and Gram stain, being repeated prior to release of the islets for transplant. Islets were deemed suitable for transplantation if they reached the appropriate release criteria defined as greater than 5000 IEQ per kg of recipient body weight, a negative Gram stain, less than 5 EU/kg endotoxin and the total tissue volume less than 10 mL as based on CIT release criteria published previously [20].

### 4.4. Recipient Patients

The patient selection criteria and outcomes of the trial have been published previously. Eligible patients had type 1 diabetes mellitus for more than 5 years and were aged between 18 and 65 years. Additionally, they had recurrent severe hypoglycaemia unawareness that required constant monitoring or regular intervention by a third party with a hyposcore as assessed by the Edmonton criteria of greater than 1000 [21]. All patients gave informed consent, and the protocol was approved by the Human Research Ethics Committee of the Western Sydney Local Health District.

### 4.5. Islet Transplantation

For the purposes of this evaluation, all islets were isolated, and all patients were transplanted at Westmead Hospital. The islets were resuspended in 120 mL of medium 199 (ThermoTrace, Melbourne, Australia) containing 5000 U heparin and 20% human albumin. Patients received a general anaesthetic, and a mini-laparotomy was performed to access a mesenteric vein. An arterial angiographic catheter was inserted and threaded into the main portal vein with the assistance of image intensification and the islets infused under gravity.

### 4.6. Data Collection and Grouping

Donor information was collected at the time of retrieval on national organ donor data sheets supplied with the organ by the organ donor agency. Additional information was also requested from the relevant organ donor agency if this was missing. Distance from islet isolation facility was calculated as-the-crow-flies and then grouped into six zones: 1 =< 50 km, 2 = 50 to 150 km, 3 = 151 to 400 km, 4 = 401 to 800 km, 5 = 801 to 1200 km and 6 = 1201 to 4000 km (Figure 1). The distances were divided into zones to incorporate the local Sydney Metro region in zone 1, the wider Sydney region in zone 2, the rest of highly populated NSW and the Australian Capital Territory in zone 3, major cities Melbourne and Brisbane in zone 4, major cities Adelaide, Hobart, and Rockhampton in zone 5 and remote areas of Australia more than 1200 km from the Isolation facility including the city of Perth some 3290 km away in zone 6.

For CIT, age and days in ICU, absolute values were used for analysis. For gender, females were assigned to group 1 and males to group 2. Retrieval team was split into group 1—our local team and group 2—all other retrieval teams.

Cause and mechanism of death were classified into CITR structured groups^13^ by agreement between two members of our team (Surgeon and Islet Operations Manager). Inotrope use was classified into 4 groups: 1 = none, 2 = normal/low (4–6 mg/100 mls at <6 mls/h), 3 = moderate (4–6 mg/100 mls at 6–10 mls/h) and 4 = high (4–6 mg/100 mls at >10 mls/h). Steroid use was assigned a 0 for none and 1 for some and C-peptide in recipients 0 for a negative isolation (<0.2 at one month) and 1 for a positive isolation (>0.2 at one month).

A positive isolation outcome was defined as ‘preparation of transplant quality islets meeting all release criteria allowing for transplantation of the islets’. A positive transplant outcome was defined as ‘recipient C-peptide >0.2 at 1 month post-transplant’.

### 4.7. Statistical Analysis

The statistical software package S-PLUS v8 was used to analyse the data. Ranges (or minimum/maximum values), means and standard deviations (SD) were calculated for all data. For non-continuous data, a value was assigned to each group for analysis (Table 1), and Chi-square and Fisher’s Exact tests were used to test significance (including Table 2). Non-parametric Spearman’s rank correlations were used for transplant outcome analysis. A *p*-value of less than 0.05 was considered significant.

## Figures and Tables

**Figure 1 metabolites-11-00360-f001:**
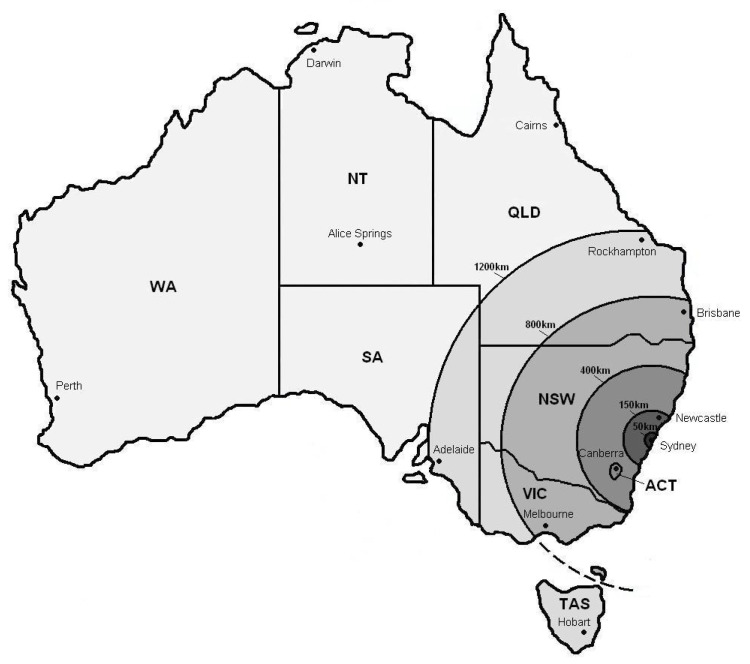
Distance zones from Westmead Hospital. Schematic map of Australia with states and major cities denoted. Distance rings (different areas of shading) are shown radiating from NPTU in Westmead Sydney, NSW.

**Figure 2 metabolites-11-00360-f002:**
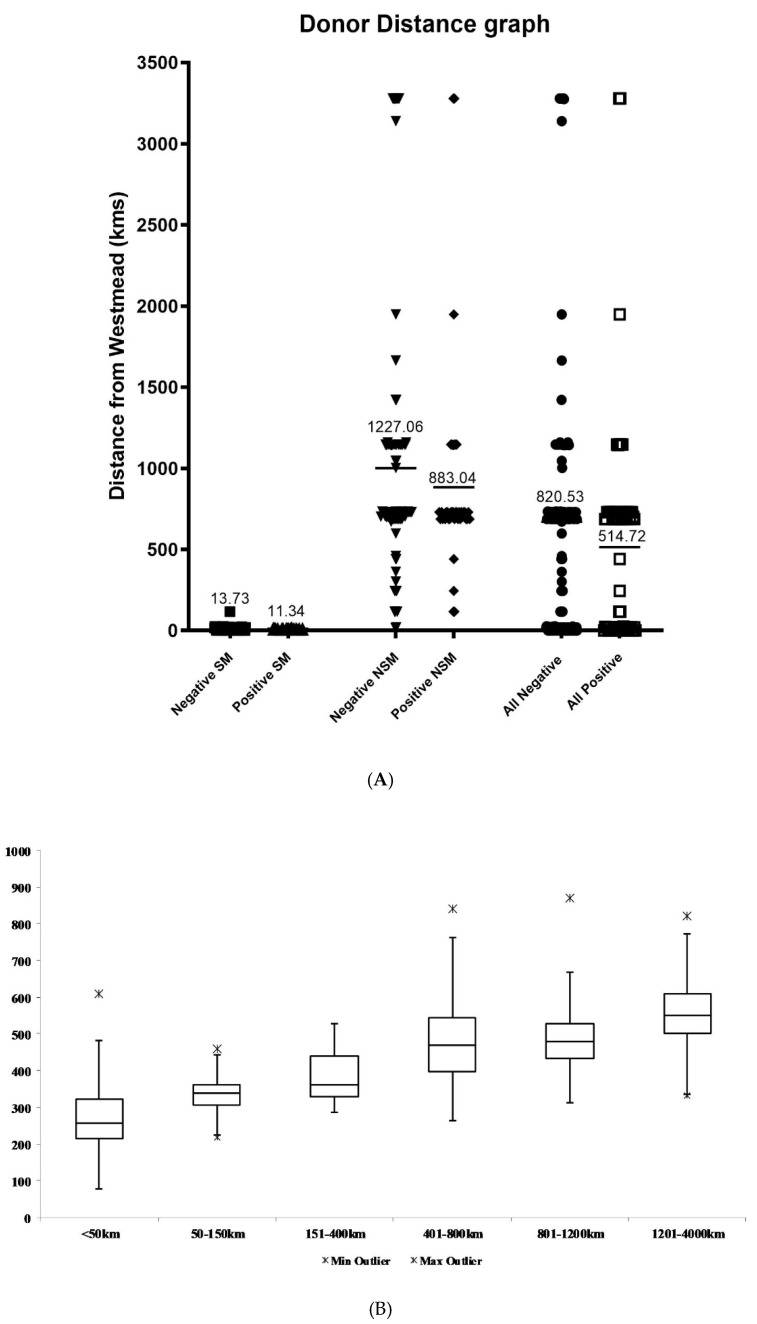
(**A**). Scatter plot of donor distance outcomes. Distance from Westmead (NPTU) in kilometers is shown for each analysis group, and the mean of each group is marked and labelled. Negative local (■ *n* = 65), positive local (▲ *n* = 30), negative non-local (▼ *n* = 129), positive non-local (◆ *n* = 41), all negative (● *n* = 194), all positive (☐ *n*= 71). (**B**). Box plot of donor distance zone vs. CIT. The CIT is shown for each distance zone, and outliers are represented by 🞶 on the graph. <50 km *n* = 71, 50–150 km *n* = 12, 151–400 km *n* = 9, 401–800 km *n* = 76, 801–1200 km *n* = 31, 1201–4000 *n* = 17. From local (zone 1) to zone 2, the mean CIT increases from 279.6 to 337.6 min which was statistically different (*p* = 0.03). There was no significant difference in the mean CIT from zone 2 to zone 3 (385.1 min) (*p* = 0.182), from zone 3 to zone 4 (474.8 min) (*p* = 0.009) and from zone 4 to zone 5 (504.9 min) (*p* = 0.171) (each zone representing a distance of 250 to 400 km). There was also a significant increase in CIT when a donor pancreas was received from zone 6 compared to zone 5 (*p* = 0.109), as zone 6 represents a distance of 2800 km further than zone 5 (mean CIT 560.6 min). (**C**). Scatter plot of CIT Outcomes. CIT in minutes is shown for each analysis group, the mean of each group is marked and labelled. Negative local (■ *n* = 67), positive local (▲ *n* = 32), negative non-local (▼ *n* = 86), positive non-local (◆ *n* = 33), all negative (● *n* = 153), all positive (☐ *n* = 65). CIT was statistically significant between local and non-local groups (*p* < 0.01).

**Figure 3 metabolites-11-00360-f003:**
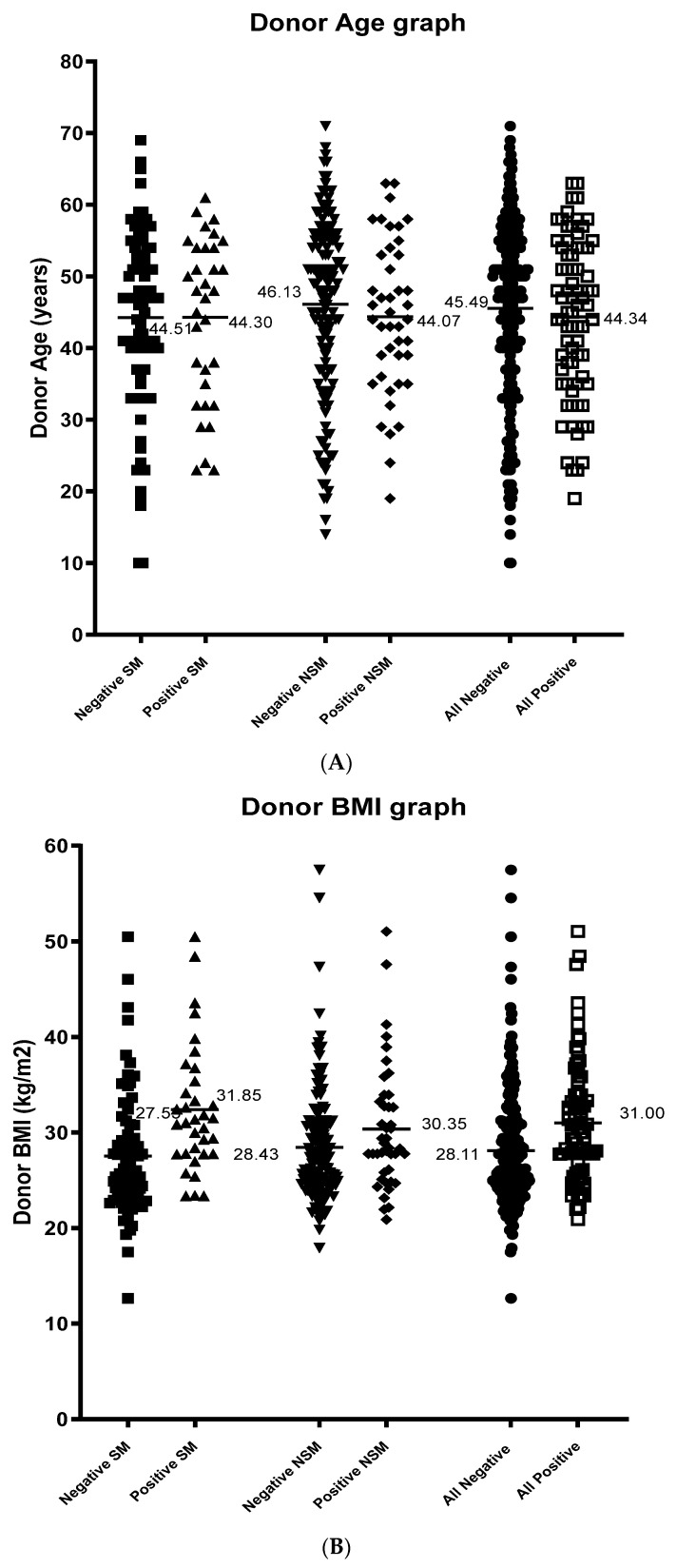
(**A**). Scatter plot of donor age outcomes. The age of the donor at time of donation in years is shown for each analysis group, the mean of each group is marked and labelled. Negative local (■ *n* = 79), positive local (▲ *n* = 33), negative non-local (▼ *n* = 142), positive non-local (◆ *n* = 43), all negative (● *n* = 221), all positive (☐ *n*= 76). (**B**). scatter plot of donor BMI outcomes. The BMI as calculated from height and weight of the patient is shown for each analysis group, the mean of each group is marked and labelled. Negative local (■ *n* = 79), positive local (▲ *n* = 33), negative non-local (▼ *n* = 142), positive non-local (◆ *n* = 43), all negative (● *n* = 221), all positive (☐ *n* = 76). BMI was statistically significant between negative and positive groups (*p* = 0.0004). (**C**). Scatter plot of donor ICU outcomes. The amount of time in days that the donor spent in ICU is shown for each analysis group, the mean of each group is marked and labelled. Negative local (■ *n* = 52), positive local (▲ *n* = 25), negative non-local (▼ *n* = 129), positive non-local (◆ *n* = 39), all negative (● *n* = 181), all positive (☐ *n* = 64).

**Table 1 metabolites-11-00360-t001:** Descriptive statistics of the data collected, grouped into donor-related factors, those that are Edmonton score donor-related factors and recipient-related factors. For non-continuous data, a number was assigned to each set of data.

	N	Min	Max	Mean	SD
**DONOR-RELATED FACTORS**					
**Outcome**	263	0.00	1.00	0.27	0.44
**Donor Distance (kms)**	263	0.00	3278.70	780.06	2195.2
**Donor Distance Zone**	263	1.00	6.00	3.09	1.73
**EDMONTON SCORE DONOR-RELATED FACTORS**					
**CIT (mins)**	215	124.00	870.00	404.35	140.21
**Donor Age**	296	10.00	71.00	45.22	13.03
**Donor BMI**	298	19.80	57.46	28.82	6.01
**Donor Gender**	308	1.00	2.00	1.49	0.49
**Donor Days in ICU**	252	1.00	13.00	3.07	2.51
**Retrieval Team**	268	1.00	2.00	1.56	0.50
**Cause of Death**	246	1.00	6.00	2.64	0.84
**Mechanism of Death**	246	1.00	12.00	7.78	1.98
**Inotrope Usage**	244	0.00	3.00	1.38	1.00
**Steroid Usage**	244	0.00	1.00	0.02	0.13
**RECIPIENT-RELATED FACTORS**					
**C-Peptide at 1 Month Post-Transplant**	23	0.00	0.91	0.28	0.23
**C-Peptide level**	73	0.00	1.00	0.55	0.36

**Table 2 metabolites-11-00360-t002:** Cause and mechanism of donor death statistics. The number of each cause or mechanism that had a negative or positive isolation outcome is compared with the percent of each total. ‘Other causes’ or ‘other mechanisms’ have the highest positive results with a significant *p* value, while those with anoxia or cardiac arrest have the least. Sharp injury and seizure as a mechanism of death have the lowest numbers. However, total numbers in these groups are very low.

**Cause of Donor Death**	**Outcome (N)**		**% Positive**	***p* Value**
	**Negative**	**Positive**		
**Anoxia/Cardiac Arrest**	22	4	15.38	0.113
**Head Trauma**	40	17	29.82	0.829
**Cerebrovascular/Stroke**	108	44	28.95	0.909
**Other**	4	5	55.56	0.124
**Total**	174	70	28.69	-
**Mechanism of Donor Death**	**Outcome (N)**		**% Positive**	***p* Value**
	**Negative**	**Positive**		
**Asphyxiation**	8	2	20.00	0.728
**Blunt Injury**	2	1	33.33	1.000
**Cardiovascular**	4	1	20.00	1.000
**Sharp Injury**	4	0	0.00	0.325
**Intracranial Haemorrhage/Stroke**	148	58	28.16	0.396
**Seizure**	1	0	0.00	1.000
**Other**	7	8	53.33	0.042
**Total**	170	70	28.69	-

**Table 3 metabolites-11-00360-t003:** Factors affecting C-peptide levels. The correlation coefficient of each factor in relation to C-peptide level as a positive transplant outcome are shown with their significance. Donor distance, CIT, donor BMI, donor days in ICU and recipient distance all showed a weak positive correlation to C-peptide levels (as one increases, C-peptide level increases).

	C-Peptide Level	
	Correlation Coefficient	Significance (2-Tailed)
**Donor Distance (kms)**	0.311	0.832
**CIT (mins)**	−0.308	0.065
**Donor Age**	0.354	0.977
**Donor BMI**	−0.171	0.770
**Donor Days in ICU**	0.135	0.659

## Data Availability

The data presented in this study are available in article.

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
