# Peer review of "Successful Islet Outcomes Using Australia-Wide Donors: A National Centre Experience"

_metabolites, 2021, doi:10.3390/metabo11060360_

Round 1
Reviewer 1 Report
The paper from Wayne Hawthorne et al. is talking about allogeneic islet Tx for recipients with T1D using deceased pancreas donors from distant locations.
Although the condition of the organ is different than one for AutoTx -- due to variables such as cause of death and length of hospital stay, which can affect islet isolation -- it is known that besides cold ischemia time (CIT) the problem is not per se the geographical distance existing between the donor site and the recipient’s place. The paper of Johnston et al. (Factors associated with islet yield and insulin independence after total pancreatectomy and islet cell autotransplantation in patients with chronic pancreatitis utilizing off-site islet isolation: Cleveland Clinic experience.J Clin Enodocrinol Metabol 100(5):1765-70. doi: 10.1210/jc.2014-4298, 2015. PMID: 25781357), for example, reaches exactly the same conclusions here presented. This dramatically reduces thenovelty of this report, even if it might be still important to indicate the best allocation of resources, which may maximize patient outcomes in the Australia’s unique situation.
Besides the novelty concern, the main problem with the paper is that it seems it was written in a hurry: I’ll indicate some points as examples where the numbers that they give in the text do not match up with the numbers that they quote previously; or they don't define exactly what they mean with "isolation outcome" since seemingly contradicts their main point which is that CIT/distance (up to their limit of 676 minutes) does not play a role in it.
It must be because they are loose with their wording and don’t make a distinction between "successful islet isolation" which they don't define and it "meets release criteria" which is standardized. Longer periods of CIT makes no difference for "successful isolation" (meaning what?) but it negatively affects "islet release criteria" which means they more often cannot be used for Tx.
(#1) Page 1 – Abstract – lines 117-119 – says 143 isolations (32 local and 111 non-local) between 2007-2014 but Page 2 – Results – lines 48-50 – shows the same 32 local and 111 non-local but says this is from 2007-2021 and includes 297 isolations. They should update abstract to include 2014-2021 and/or update results to indicate how many of the additional (297-143) isolations are local and nonlocal.
(#2) Page 2- results – line 52 says that distance (and presumably CIT) from donor to isolation facility had no impact on islet isolation outcome but lines 55-57 show statistically significant difference in whether or not islet release criteria were met. Meeting or failing release criteria are arguable the most important isolation outcome (i.e., will the patient get islet Tx or not) so they might want to define “positive or negative” islet isolation outcome before making this statement. I am assuming it is a certain # of islets or IEQ/kg but they don’t define it here. Islet ISOLATION outcomes are not the same as islet TX outcomes.
(#3) Page 5 – Other factors effecting isolation outcome – lines 97-109 – the values for age, sex, BMI, time in ICU are not the same as in the abstract probably reflecting the missing data in point #1.
(#4) Page 8 – Discussion – lines 158-160 – contradicts point #2 (Page 2 says “distance from the isolation facility provided no significant impact on the islet isolation outcomes” and “CIT [cold ischemia time] correlated closely with the distance from isolation facility” while Page 8 now says “This [substantial differences in CIT time] did impact on the chances of achieving a successful isolation. However, provided the release criteria were met there were no differences in transplant outcomes between local and non-local donors.” This statement on page 8 is correct leading me to believe that the previous statements are less accurate and should be reworded accordingly.
(#5) Page 10 – Materials and Methods – lines 228-237 - Release Criteria is identified but nowhere is it stated what they mean by “positive or negative islet isolation outcome” although they talk about it repeatedly as something different and independent of the release criteria.
(#6) Graphs (Figure 2 – page 3, Figure 4 – page 4, Figure 5 – page 5, Figure 6 – page 6, Figure 7 - page 7) use terminology “SM” and “NSM.” Based on the shapes used in the plotting of data, “SM” means local and “NSM” is non-local but it would be helpful to have it spelled out clearly in the legend. It is such a small detail that it might be mentioned somewhere in the text but I did not see it.
Author Response
We thank the reviewer for their comments and constructive suggestions in relation to the attention to detail of the manuscript. We have made all the suggested changes and they are detailed below.
Although the condition of the organ is different than one for AutoTx due to variables such as cause of death and length of hospital stay, which can affect islet isolation it is known that besides cold ischemia time (CIT) the problem is not per se the geographical distance existing between the donor site and the recipient’s place. The paper of Johnston et al. (Factors associated with islet yield and insulin independence after total pancreatectomy and islet cell autotransplantation in patients with chronic pancreatitis utilizing off-site islet isolation: Cleveland Clinic experience.J Clin Enodocrinol Metabol 100(5):1765-70. doi: 10.1210/jc.2014-4298, 2015. PMID: 25781357), for example, reaches exactly the same conclusions here presented. This dramatically reduces the novelty of this report, even if it might be still important to indicate the best allocation of resources, which may maximize patient outcomes in the Australia’s unique situation.
We acknowledge the reviewer’s point that the paper from Johnston et al. reached similar conclusions that many variables can impact isolation outcomes along with transplantation outcomes. However, their paper is purely based upon the auto isolation experience from one centre in the USA, where the only mention of any factor related to CIT was “Total time from pancreatectomy to return of the islets to Cleveland was around 8–9 hours.” There were NO distances mentioned nor any correlation with any such variables performed as in our manuscript. Their paper is from only 36 patients compared to 297. All of their donor/recipient patients underwent controlled pancreatectomy where all patient characteristics are fully assessed, and the surgical procedure occurs in a controlled environment at a precisely scheduled time, all pancreatic surgeries were performed by a single surgeon (R.M.W.) which is nothing like what occurs in the allo setting. As such it is not possible to compare Auto with Allo for such situations, nor the many differing variables associated with allo donors. It is unfair to suggest that these settings are the same or the many variables the same. Additionally, their manuscript is for only one distant isolation center where the time is the identical for every isolation and transport, they state 8-9hrs and provide no other information rather just an 8-9hr time point. This provides no insight into any travel time, nor does it show any variance in relation to distance, CIT, retrieval logistics, nor any donor variables such as are seen in allo organ donation.
At our centre we have also undertaken a significant number of total pancreatectomies with islet isolation and subsequent autotransplantation and we have deliberately excluded these data from this manuscript as they are not the same nor can be compared for the many donor variables that are seen in allo islet organ donors over such a vast continent as what Australia is with its many different centres and teams associated with the organ donor retrieval
Besides the novelty concern, the main problem with the paper is that it seems it was written in a hurry: I’ll indicate some points as examples where the numbers that they give in the text do not match up with the numbers that they quote previously; or they don't define exactly what they mean with "isolation outcome" since seemingly contradicts their main point which is that CIT/distance (up to their limit of 676 minutes) does not play a role in it.
We apologise for the significant oversight, and the reviewer’s assumption the manuscript was written in a hurry. This was not the case it was an unfortunate oversight in submission. Dr Mun a post doc in the lab was responsible for the final version submission and upload of the manuscript. Unfortunately, he uploaded the incorrect version of the manuscript on his last day in his job. He has subsequently left our group and this unfortunate oversight has now been corrected. We hope that we have addressed the reviewers overarching concerns due to this oversight.
We additionally have made significant changes throughout the manuscript as suggested by both reviewers. Further to this we now more clearly define “isolation outcome” It is defined as “A positive isolation outcome was defined as ‘preparation of transplant quality islets meeting all release criteria allowing for transplantation of the islets’.”
It must be because they are loose with their wording and don’t make a distinction between "successful islet isolation" which they don't define and it "meets release criteria" which is standardized. Longer periods of CIT makes no difference for "successful isolation" (meaning what?) but it negatively affects "islet release criteria" which means they more often cannot be used for Tx.
We thank the reviewer for their help with improving the manuscript. We agree with them that the definitions could be tighter and as such wed have addressed this throughout the manuscript.
This includes the methods and conclusions where we now make the following additional statements and better definitions;
“A positive isolation outcome was defined as ‘preparation of transplant quality islets meeting all release criteria allowing for transplantation of the islets’.”
“This did impact on the chances of achieving a positive isolation resulting in reaching release criteria. However, provided the release criteria were met there were no differences in transplant outcomes between local and non-local donors.”
“In this setting CIT is a good surrogate for distance travelled but only when taken in the right context and in relation to the exceptionally long distances travelled across Australia that can only occur by air travel. CIT has been shown to have an adverse outcome on islet isolation in both single centre and registry studies.”
“The BMI of a donor was also significant in determining a positive isolation outcome meeting release criteria and thus we must continue to consider this when deciding whether we should perform a donor pancreas isolation.”
(#1) Page 1 – Abstract – lines 117-119 – says 143 isolations (32 local and 111 non-local) between 2007-2014 but Page 2 – Results – lines 48-50 – shows the same 32 local and 111 non-local but says this is from 2007-2021 and includes 297 isolations. They should update abstract to include 2014-2021 and/or update results to indicate how many of the additional (297-143) isolations are local and nonlocal.
These issues have been addressed by uploading the correct version of the paper with the updates already done as suggested by the reviewer. We show the changes associated with what was updated to the Abstract and remainder of the text.
(#2) Page 2- results – line 52 says that distance (and presumably CIT) from donor to isolation facility had no impact on islet isolation outcome but lines 55-57 show statistically significant difference in whether or not islet release criteria were met. Meeting or failing release criteria are arguable the most important isolation outcome (i.e., will the patient get islet Tx or not) so they might want to define “positive or negative” islet isolation outcome before making this statement. I am assuming it is a certain # of islets or IEQ/kg but they don’t define it here. Islet ISOLATION outcomes are not the same as islet TX outcomes.
The reviewer is correct, and we have rephrased what we say in this section to consider the differences rather than looking at the means of all data.
We now state “The distance from the isolation facility provided no significant impact on the islet isolation outcome (the closer local donors provided a positive outcome at 11.34 km ± 10.22 vs 883.04 km ± 625.45 for a negative outcome from non-local donors, p=<0.01) (Fig. 2A). The chances of obtaining an isolation outcome that achieved release criteria was greater from local donors with a positive outcome occurring 32% of the time from local donors as compared to only 24% of isolations from non-local donors (p<0.01).”
In the methods section we clearly define the differences between islet isolation outcomes and transplant outcomes and we also had the definition of what release criteria were for transplant stating “Islets were deemed suitable for transplantation if they reached the appropriate release criteria defined as – greater than 5,000 IEQ per kg of recipient body weight”.
(#3) Page 5 – Other factors effecting isolation outcome – lines 97-109 – the values for age, sex, BMI, time in ICU are not the same as in the abstract probably reflecting the missing data in point #1.
The reviewer is correct, and we have addressed these as per point 1.
(#4) Page 8 – Discussion – lines 158-160 – contradicts point #2 (Page 2 says “distance from the isolation facility provided no significant impact on the islet isolation outcomes” and “CIT [cold ischemia time] correlated closely with the distance from isolation facility” while Page 8 now says “This [substantial differences in CIT time] did impact on the chances of achieving a successful isolation. However, provided the release criteria were met there were no differences in transplant outcomes between local and non-local donors.” This statement on page 8 is correct leading me to believe that the previous statements are less accurate and should be reworded accordingly.
The reviewer is correct, and we have addressed these as above.
(#5) Page 10 – Materials and Methods – lines 228-237 - Release Criteria is identified but nowhere is it stated what they mean by “positive or negative islet isolation outcome” although they talk about it repeatedly as something different and independent of the release criteria.
We have addressed these as above and this includes the following additional statement and better definitions throughout the text as described previously and is “A positive isolation outcome was defined as ‘preparation of transplant quality islets meeting all release criteria allowing for transplantation of the islets’.”
(#6) Graphs (Figure 2 – page 3, Figure 4 – page 4, Figure 5 – page 5, Figure 6 – page 6, Figure 7 - page 7) use terminology “SM” and “NSM.” Based on the shapes used in the plotting of data, “SM” means local and “NSM” is non-local but it would be helpful to have it spelled out clearly in the legend. It is such a small detail that it might be mentioned somewhere in the text but I did not see it.
As the reviewer has pointed out the terms SM and NSM in the figures were not defined, as such we have changed these terms in all figures to be ‘positive or negative’ to reflect the reminder of the text so it is the same throughout.
Reviewer 2 Report
Main Comments
The authors present data showing good outcomes for islet isolation at 2 national centers in Australia. The analysis is fair and fairly thorough but could be tightened.
- There are far too many figures. Please combine some of your figures into multi-panel figures. E.g. Figures 2/3/4 may be combined as could figures 5/6/7.
- “Negative” vs. “Positive” outcome is not well defined and a bit awkward. Consider using another term. Also, SM and NSM are not defined either. Please provide information on these acronyms.
Specific Comments
- The last sentence of the first paragraph in results using odd units. Please give a percentage successful outcomes for isolations (I think that they would be 32% and 24% but am not entirely sure what your intention is).
- The sentence “How-64 ever regardless of the region there was a negative association between CIT and chances of 65 a positive isolation outcome” deserves statistical support. A simple linear regression may be appropriate.
- Table 2 should be analyzed by a chi2 or fisher exact test (depending on the number in each cell) in order to determine if any COD is over-represented in one group or the other.
- The sentence “CIT is a good surrogate for distance travelled and this has been shown to have an adverse outcome on islet isolation in both single centre and registry studies.” Seems to have the causality in reverse. Isn’t in the cold ischemia time and not the distance travelled that ultimately causes the problem? E.g. One might fly between cities in an hour or drive 6 hours, the distance would be the same but the effect on the organ should be greater in the driving group because of greater CIT?
Author Response
We thank Reviewer 2 for their very fair comments and constructive suggestions in relation to the manuscript. We have made all the suggested changes and they are detailed below.
- There are far too many figures. Please combine some of your figures into multi-panel figures. E.g. Figures 2/3/4 may be combined as could figures 5/6/7.
We have changed the figures as requested and now figures 2/3/4 are figure 2a, b and c, figures 5/6/7 are now figure 3a, b and c. The figure legend has also been changed to reflect this along with the text throughout the manuscript to reflect the changes to the numbering.
- “Negative” vs. “Positive” outcome is not well defined and a bit awkward. Consider using another term. Also, SM and NSM are not defined either. Please provide information on these acronyms.
As per reviewer one’s similar comment we have amended the definition to now more clearly state; A positive isolation outcome was defined as ‘preparation of transplant quality and meeting all release criteria’. A positive transplant outcome was defined as ‘recipient C-peptide >0.2 at 1 month post-transplant’.
As the reviewer has pointed out the terms SM and NSM in the figures were not defined, as such we have changed these terms in all figures to be ‘positive or negative’
Additionally the following definitions for the acronym is in the Abbreviations list.
SD Standard deviation
Specific Comments
- The last sentence of the first paragraph in results using odd units. Please give a percentage successful outcomes for isolations (I think that they would be 32% and 24% but am not entirely sure what your intention is).
The intension in this sentence was to provide an outcome measure for the number of times an isolation needs to be done to result in a transplant, this is very important in relation to funding the overall isolation and transplantation program. Funding agencies request this information to understand the ratio, as such we have changed the units here to % to help better define this. It now reads as suggested “A positive outcome occurred 32% of the time from local donors as compared to only 24% of isolations from non-local donors (p<0.01).”
- The sentence “How-64 ever regardless of the region there was a negative association between CIT and chances of 65 a positive isolation outcome” deserves statistical support. A simple linear regression may be appropriate.
We agree this deserved statistical support and so have provided a pvalue for justification of the wording in the sentence being an association rather than statistical significance.
Table 2 should be analyzed by a chi2 or fisher exact test (depending on the number in each cell) in order to determine if any COD is over-represented in one group or the other.
We have redone the statistics in Table 2. and have added p values into the table, as suggested we redid the stats using Chi2 or fisher exact tests depending on the cell. We also inserted reference to this in the methods section. Importantly no results analysed changed.
- The sentence “CIT is a good surrogate for distance travelled and this has been shown to have an adverse outcome on islet isolation in both single centre and registry studies.” Seems to have the causality in reverse. Isn’t in the cold ischemia time and not the distance travelled that ultimately causes the problem? E.g. One might fly between cities in an hour or drive 6 hours, the distance would be the same but the effect on the organ should be greater in the driving group because of greater CIT?
We agree that there is definite difference between the fly times and drive times resulting in differing CIT, and as such to strengthen this discussion point, we have added the wording “when taken in the right context and in relation to the exceptionally long distances travelled across Australia that can only occur by air travel where”.
We have also added the following text in the discussion; “In most instances though the careful planning of organ donor retrieval surgical times around flight times and urgency to meet assigned flights made significant differences to flight times as seen between zone 6 and those undertaken in zone 4 where similar urgency to make flight times was not as critical or there were significant distances driven from a distant regional hospital to make a domestic flight interstate.”
We hope that the significant number of changes to the manuscript have addressed all of the issues the reviewers have with it and it is now a significantly improved version suitable for publication.